# Under Blue Light Treatment, OsCSN2 Regulates the Phenotype of Rice Seedlings Through the GA Signaling Pathway

**DOI:** 10.3390/plants14132015

**Published:** 2025-07-01

**Authors:** Xinhai Yu, Tongtong Jiao, Changfeng Liu, Hexin Zhang, Yanxi Liu, Chunyu Zhang, Ming Wu, Liquan Guo

**Affiliations:** 1College of Life Sciences, Jilin Agricultural University, Changchun 130118, China; 15164357527@163.com (X.Y.); jtt1163933169@163.com (T.J.); 13019596383@163.com (C.L.); zhx03101212@163.com (H.Z.); lyx309660408@163.com (Y.L.); 2Jilin Institute of Biology, Changchun 130012, China; 3College of Food and Biotechnology, Changchun Polytechnic, Changchun 130033, China; zhangchunyu2011917@163.com

**Keywords:** blue light, COP9 signaling complex, OsCSN2

## Abstract

Blue light is a significant environmental cue influencing plant photomorphogenesis and regulating plant growth and development. The COP9 signaling complex (CSN), a multi-subunit protein complex, plays a pivotal role in regulating photomorphogenesis, with CSN2 being identified as a key subunit essential for the assembly and function of the CSN. This study investigated the role of OsCSN2 in rice under blue-light conditions. Utilizing *OsCSN2* knockout (KO) mutant plants and transgenic overexpression (OE) lines for wild-type (WT) and mutated versions of OsCSN2, we observed significant suppression of the overall seedling phenotype under blue light, indicating that OsCSN2 acts as a negative regulator of blue light-mediated morphogenesis. Further analysis revealed that exogenous application of gibberellin (GA_3_) and the GA synthesis inhibitor paclobutrazol (PAC) modulated seedling elongation in response to blue light, particularly affecting plant height, coleoptile, and first incomplete leaf length without altering root growth. This suggests that OsCSN2 mediates the inhibitory effects of blue light on aboveground development through the gibberellin signaling pathway. On day 9, the analyses of endogenous GA_3_ levels combined with Western blotting (WB) and quantitative real-time PCR (qRT-PCR) revealed that OsCSN2 senses blue light signals through cryptochrome 2 (CRY2), influences the expression of *COP1* and *BBX14*, and highlights its role in the photoreceptive signaling pathway. This regulation ultimately influences the degradation of SLR1 within the GA signaling pathway, affecting rice seedling growth and development. Our findings also highlight the differential roles of OsCSN1 and OsCSN2 within the CSN in modulating rice seedling photomorphogenesis, thereby providing new insights into the intricate regulatory mechanisms governing plant responses to blue light.

## 1. Introduction

The COP9 signaling complex (CSN) is a highly conserved protein complex. It was first found in *Arabidopsis thaliana* (*Arabidopsis thaliana* (L.) Heynh.) to be an essential regulator that negatively regulates photomorphogenesis [1]. It also participates in plant floral development, defense, and hormone responses [2,3]. Later, it was also found to be involved in the developmental functions of various eukaryotic cells. In plants, it usually comprises eight subunits, which are named CSN1 to CSN8 [1]. CSN2 is the second subunit of the CSN, and it has different names in different species. It is known as *Sgn2*, *TRIP15*, and *hAlien* in humans, *COPS2* in mice, and *FUS12* in *Arabidopsis* [3]. In the COP9 signaling complex, CSN2 is considered to be a linker and plays an essential role in the assembly of the complex and substrate recognition [4]. Previous studies revealed that CSN2 in mammals and plants contains a proteasome, COP9, initiation Factor 3 (PCI) domain. The PCI domain contains a pure β-helical domain and is usually located at the C-terminus; it is also present in 26S proteasomes. The lid subunit and eukaryotic translation initiation Factor 3 (eIF3) have catalytic activity [5,6].

In plant physiological processes, protein ubiquitination is a critical post-translational modification of proteins in the signaling regulation process [1]. Ubiquitination and proteasome-mediated protein degradation play essential roles in the growth and development of organisms [1,7]. The 26S proteasome is a giant complex that recognizes ubiquitinated substrates. Each subunit in the CSN is homologous to one of the subunits of the bona fide lid subcomplex of the 26S proteasome. Some researchers have speculated that CSN may replace the bona fide lid subcomplex to affect the proteolytic activities of the 26S proteasome and ubiquitin ligases [1,8]. CSN also acts as a regulator of the Cullin-RING E3s ligases (CRLs), which regulate protein degradation through deoxygenation, thereby affecting various developmental processes in plants. In the SCF subfamily, which includes E3 ligases, the CSN functions as a regulator of Skp-Cullin-F box (SCF) ubiquitin ligases [3,9]. CSN can regulate the activity of many ubiquitin ligases, such as constitutive photomorphogenesis protein 1 (COP1), and both are negative regulators of photomorphogenesis; however, the specific interaction between them remains to be studied further [10]. It is also crucial in the research of the food crop rice (*Oryza sativa* L.).

Rice is an essential food crop in China, and ensuring the yield of rice is an essential task of economic development in China. Rice is a photoautotrophic organism that uses the absorption of light energy to manufacture organic matter. Light factors are very essential in the whole growth cycle of rice [11]. Rice organic matter accumulation mainly originates from photosynthesis, but the light energy utilization efficiency is very low. Improving the light energy utilization efficiency is also one of the hotspots at present. Studies of light-regulated chlorophyll synthesis have focused on the transition process of seedlings from heterotrophic growth to autotrophic growth under light. Blue light is an essential light source. Blue light can increase the photosynthetic capacity, which is beneficial for controlling plant height. Blue light can affect the photomorphogenesis process of plants and can also increase the chlorophyll a/b ratio [12]. Blue light also has relevant effects on the physiology of rice seedlings. Blue light also affects elongation in tomatoes. Moreover, blue light can stimulate the photosynthetic potential in cucumbers. Increasing the proportion of blue light can increase the photosynthetic capacity of leaves and can improve the accumulation of chlorophyll in hemp in industrial applications; it increased stem diameter, shoot fresh weight, dry weight, and so on [13].

Blue light receptors that sense blue light include cryptochrome (CRY) and phototropin (PHOT), both of which absorb blue light at wavelengths ranging from 315 to 500 nm [14]. CRY exists in the form of a dimer and plays an essential role in biological functions, including the inhibition of hypocotyl elongation by blue light, mainly through the regulation of gene expression [15,16]. Rice contains three CRY family members: OsCRY1a, OsCRY1b, and OsCRY2. CRY participates in various biological responses to blue light. Overexpression of OsCRY1 can lead to significant inhibition of leaves and coleoptiles under blue light [17], inhibition of hypocotyl elongation, and anthocyanin accumulation in *Arabidopsis* [18]. Plants exhibit phototropism in response to blue light, and PHOT1 is the main photoreceptor involved in the phototropic response [19]. In rice, PHOT has three members, OsPHOT1a, OsPHOT1b, and OsPHOT2, in addition to the presence of blue light receptors such as ZEITLUPE (ZTL) under blue light [15,20]. COP1, which was first screened and characterized in plants for regulation by photoreceptors, is known as a “star protein”, a negative regulator of photomorphogenesis, and a single-protein finger-like E3 ubiquitin ligase [21]. COP1 promotes the degradation of positive regulators of photomorphogenesis such as ELONGATED HYPOCOTYL 5 (HY5), which is driven by the 26S proteasome, and also assembles with CUL4 ubiquitin ligase to form a complex-type CUL4 E3 ubiquitin ligase [21]. Previously reported in the literature, yeast two-hybrid experiments revealed that OsCRY1b and OsCOP1 interact and that both OsCRY1b and OsCOP1 are localized in the nucleus [18]. In rice and *Arabidopsis*, CRYs inhibit E3 ubiquitin ligases by forming complexes with SPA1 and COP1 in a blue light-dependent manner [16]. The PIFs protein family is an essential member of light signal transduction, and its activity is affected not only by light but also by hormones, such as gibberellic acid and brassinosteroids [22]. PIF4 and PIF5 regulate the blue light response in *Arabidopsis* through direct interactions with cryptochrome and target genes [16]. The transcription of AtHY5 plays an essential hub role in the regulation of photomorphogenesis. Its homologs in rice are *OsbZIP18* and *OsbZIP48* [23,24]. The phenotype of the *Arabidopsis athy5* mutants was restored after *OsbZIP48* gene transfer. However, the height of *AtHY5*-overexpressing Arabidopsis plants was the same as that of wild-type plants. Sometimes, the plant height of the mutants was even slightly lower than that of the wild type [23]. Under light treatment, the photoreceptor first inhibits COP1 and then HY5, and many proteins are involved in this process of photoreceptor signaling, such as the BBX protein, which is involved in the regulation of photomorphogenesis in seedlings. OsBBX14 is involved in blue light regulation, and OsBBX14 physically interacts with the OsCRY2 protein [25]. Blue light is involved in various signaling pathways and physiological changes, and the light signal transduction pathway is a key research target.

Phytohormones play essential physiological and biochemical roles in the growth of plants, such as gibberellin (GA) and abscisic acid (ABA). GA can promote seed germination and stem elongation [26]. ABA has antagonistic effects on these processes. Previous studies have preliminarily explored the regulatory effect of *OsCSN1* mutants on the addition of exogenous hormone GA under blue light, and found that OsCSN1 may mediate the degradation of SLR1 through CUL4 under blue light, affecting the GA signaling pathway to regulate the growth and development of rice seedlings. Both OsCSN1 and OsCSN2 are important constituent subunits of the COP9 signaling complex. This study aims to investigate whether the effects of *OsCSN2* mutants on blue light and exogenous hormone GA differ from those of OsCSN1. All knockout and overexpression mutant seeds used in this experiment were obtained from our laboratory. Point mutants were predicted for the OsCSN2 protein sequence and found that the probability of lysine ubiquitination at sites 64, 67, and 104 of OsCSN2 was as high as 90%, and the lysine (K) located at these three different sites was changed to glutamic acid (E) to make it deubiquitinated, a secondary structure comparison between the point mutants and OsCSN2 revealed that each point mutant had an extra β-sheet [27]. This may change its original ubiquitination and affect the degradation of proteins by the ubiquitin–proteasome system. The OsCSN2-related signaling pathways were studied via blue light as well as the addition of the exogenous hormone gibberellin (GA_3_) and the gibberellin synthesis inhibitor paclobutrazol (PAC). Previously, the gibberellin signaling pathway was reported to sense gibberellin through the GA receptor GID1 to inhibit the degradation of the DELLA protein in the GA signaling pathway [27]. DELLA proteins play a repressive role in GA signaling, and DELLA proteins interact with transcription factors to inhibit the DNA-binding activity of transcription factors and thus repress the expression of target genes; there is only one DELLA protein in rice, namely SLR1 [17]. In rice, OsCSN1 can regulate the nuclear localization of COP1 through the COP9 signaling complex and then degrade SLR1 through the E3 ligase of CUL4 [17], thereby regulating the growth and development of rice seedlings.

In recent years, more studies on CSN2 in animals and *Arabidopsis* have been reported, but fewer studies have been reported in rice. Studies on the involvement of CSN2 in the regulation of seedling growth and development and relevant studies on blue light signaling pathways are rare. It remains to be explored whether the OsCSN1 subunit has the same function in the blue light and rice GA signaling pathways, which is highly essential for elucidating the regulatory mechanisms of rice growth and development.

## 2. Results

Wild-type rice Nipponbare (WT), *oscsn2-1*, *oscsn2-2*, *OsCSN2-OE*, *OsCSN2K64E-OE*, *OsCSN2K67E-OE*, and *OsCSN2K104E-OE* plants were grown in media for 9 days, and the rice seedling phenotype was determined. Analysis of the rice seedling phenotypic data revealed that the aboveground parts of rice seedlings were more sensitive to the GA signaling pathway, which was not apparent in the roots. The endogenous hormone contents were determined in different parts of the aboveground 9-day-old rice seedlings. Analysis of the changes in the phenotype of the rice seedlings led to the identification of a preliminary signaling pathway based on the relative expression levels of proteins and genes.

### 2.1. OsCSN2 Can Negatively Regulate the Inhibitory Effect of Blue Light on the Height of Rice Plants Through the GA Signaling Pathway

Under natural light conditions, *oscsn2-1* and *oscsn2-2* were significantly higher than the WT. Among the *OsCSN2* point mutants, the plant heights of *OsCSN2K64E-OE*, *OsCSN2K67E-OE*, and *OsCSN2K104E-OE* were greater than that of *OsCSN2-OE*, but those of *OsCSN2-OE* were shorter than those of the WT (Figure 1A,B). These results suggest that OsCSN2 has an inhibitory effect on plant height elongation in rice. Under blue light irradiation conditions, the plant heights of the WT and *OsCSN2* mutants decreased overall and the plant heights of the mutants were all shorter than the WT plant height, but the plant heights of *oscsn2-1* and *oscsn2-2* were significantly greater than that of *OsCSN2-OE* (Figure 1A,B); blue light significantly inhibited the increase in plant height. In addition, the endogenous hormone GA_3_ was measured in the stems of the rice plants. The results revealed that the overall hormone levels decreased significantly, and those of *OsCSN2-OE* were the lowest (Figure 2A). OsCSN2 may be inhibited by affecting the expression of the endogenous hormone GA_3_ through blue light. OsCSN2 is a negative regulator of blue light morphogenesis. Under co-treatment with blue light and the exogenous hormone GA_3_, *oscsn2-1*, *oscsn2-2*, and *OsCSN2-OE* restored plant height under natural light conditions (Figure 1A,B), and the expression of the endogenous hormone GA_3_ also increased significantly (Figure 2A). In particular, the heights of the plants in the *OsCSN2K67E-OE* and *OsCSN2K104E-OE* groups were significantly greater than those in the natural light group (Figure 1A,B). This demonstrated that the application of the exogenous hormone GA_3_ also promoted endogenous hormone levels in rice seedlings. OsCSN2 may rescue the inhibitory effect of blue light on the height of mutant plants, possibly through the GA signaling pathway. The changes in the *OsCSN2* point mutants may be related to the ubiquitylation of the point mutants, and the responses to GA_3_ are different, thus regulating the growth trend. After the addition of the exogenous hormone PAC, the plant height of all the rice seedlings was inhibited, and *OsCSN2-OE* was the most sensitive (Figure 1A,B). The endogenous hormone GA_3_ was also significantly inhibited (Figure 2A).

### 2.2. Exogenous GA_3_ Can Regulate the Inhibitory Effect of Blue Light on the Coleoptile and the First Incomplete Leaf of OsCSN2 Rice Seedlings

Under natural light, the coleoptile length of all the mutants was longer than that of the WT, especially the coleoptile lengths of *oscsn2-1* and *OsCSN2K64E-OE* (Figure 1A,C). These results suggest that OsCSN2 can promote the elongation of coleoptiles in rice seedlings. The coleoptile shortening of all the mutants was inhibited under blue light (Figure 1A,C), and the content of the endogenous hormone GA_3_ was also significantly suppressed and decreased (Figure 2B). Under the combined action of blue light and the exogenous hormone PAC, the WT and all the mutants were more significantly inhibited, the coleoptiles of *OsCSN2-OE* and *OsCSN2K64E-OE* were significantly shortened (Figure 1A,C), and *OsCSN2-OE* endogenous hormone content also showed lower levels (Figure 2B). Under the combined action of blue light and the exogenous hormone GA_3_, the overall trend of the coleoptiles was promoted (Figure 1A,C). In particular, the coleoptiles of the *OsCSN2K64E-OE* plants were longer than those of the WT plants. The endogenous hormone GA_3_ also showed the highest levels (Figure 2B); the addition of the exogenous hormone GA_3_ reversed the inhibitory effect of blue light on *OsCSN2* coleoptile elongation, with *OsCSN2K64E-OE* showing the greatest sensitivity. The ubiquitination sites may affect the different performances of the mutants in the GA signaling pathway.

Under natural light, there was no significant difference in the number of incomplete first leaves between the WT and mutant plants (Figure 1A,D). Blue light treatment inhibited the elongation of the first incomplete leaf. *OsCSN2* point mutants were shorter than the WT, *oscsn2-1* and *oscsn2-2* were greater than the WT, and the first incomplete leaves of both the *OsCSN2-OE* and the WT were similar (Figure 1A,D). Similarly, the difference in the levels of the endogenous hormone GA_3_ was not significant (Figure 2C). Under the combined action of blue light and the exogenous hormone PAC, the incomplete first leaves of the WT and mutant plants were more significantly suppressed (Figure 1A,D), and the endogenous hormone contents were also significantly suppressed (Figure 2C). However, under the joint action of blue light and the exogenous hormone GA_3_, the overall length of the first incomplete leaf was significantly increased to the extent that it was approximately two times longer than that under blue light and one time longer than that under natural light, especially *oscsn2-1*, so the absence of OsCSN2 could promote the incomplete first leaf of rice seedlings (Figure 1A,D) and significantly increase the content of the endogenous hormone GA_3_ (Figure 2C). These results demonstrated that GA_3_ could regulate the inhibition of the incomplete first leaf of *OsCSN2* seedlings under blue light.

### 2.3. OsCSN2 Negatively Regulates Root Length Elongation in Rice Seedlings Under Blue Light

Under natural light, the roots of *oscsn2-1* and *oscsn2-2* were significantly longer than those of the WT, and the root length of *OsCSN2-OE* was significantly shorter than that of the WT (Figure 3A,B). Therefore, OsCSN2 inhibits root length elongation in rice. Blue light negatively regulated the elongation of the roots of the WT and *OsCSN2* seedlings, especially the *OsCSN2-OE* roots (Figure 3A,B). After the addition of the exogenous hormone PAC under blue light, all the mutants were inhibited, particularly the complete inhibition of *OsCSN2-OE* root length (Figure 3A,B). After the addition of the exogenous hormone GA_3_ under blue light, the roots of both the *OsCSN2-OE* and the *OsCSN2K67E-OE* plants significantly grew, whereas the lengths of the roots of the *oscsn2-1*, *oscsn2-2*, and *OsCSN2K64E-OE* plants did not significantly change (Figure 3A,B). In summary, OsCSN2 may negatively regulate the elongation of the rice seedling root system under blue light, and OsCSN2 may rescue the inhibition of root length through the GA signaling pathway. However, it does not play a main regulatory role and may regulate OsCSN2 root development through other signaling pathways.

### 2.4. Under Blue Light, OsCSN2 May Regulate Shoot Development in Rice Seedlings via SLR1 Degradation in the GA Pathway

Real-time fluorescence quantitative PCR and WB experiments were performed on rice seedlings grown for 9 days. Under natural light, the expression level of the OsSLR1 protein in *oscsn2-1* and *oscsn2-2* was lower than that in *OsCSN2-OE* (Figure 4A). Under blue light, the expression level of the OsSLR1 protein was significantly inhibited in all the *OsCSN2-OE* mutants, especially *OsCSN2-OE* (Figure 4B), as was the mRNA level (Figure 5B). The level of the endogenous hormone GA_3_ also decreased significantly. (Figure 2A–C). Therefore, blue light inhibits the synthesis of the endogenous hormone GA_3_ and affects the elongation of the aboveground part of rice seedlings, which was most obviously inhibited in the *CSN2* overexpression mutant lines. After synergistic treatment with blue light and the exogenous hormone GA_3_, *OsCSN2-OE* presented significantly greater levels of OsSLR1 protein than the WT did, while the levels of *oscsn2-1* and *oscsn2-2* were significantly lower than those of the WT (Figure 4C). This suggests that OsCSN2 can regulate the aboveground growth of rice seedlings by regulating the content of the endogenous hormone GA_3_ and thereby affecting the degradation of SLR1. On the other hand, the expression level of the OsSLR1 protein in all the point mutants was significantly lower than that in *OsCSN2-OE*. In particular, the expression of *OsCSN2K67E-OE* was the lowest, and its aboveground parts were also significantly promoted (Figure 4C), and the alteration of the 67th ubiquitination site may be more positive for OsCSN2 in response to the GA signaling pathway. After synergistic treatment with blue light and the exogenous hormone PAC, the protein and gene expression levels of OsSLR1 significantly increased in the *OsCSN2-OE*, *OsCSN2K67E-OE*, and *OsCSN2K104E-OE* groups compared with those in the WT group (Figure 4D), whereas the expression levels of *oscsn2-1* and *oscsn2-2* also tended to increase (Figure 5D), and all the phenotypes of the rice seedlings were indeed significantly suppressed. OsABI5 protein expression levels and gene expression levels after synergic treatment with blue light and the exogenous hormone GA_3_ were affected; the expression level of the OsABI5 in *OsCSN2-OE* was significantly greater than that in WT, and the expression levels of *oscsn2-1* and *oscsn2-2* were significantly lower than those in WT (Figure 4C). The changes under the different treatments were not significant (Figure 4A,B,D). OsABI5 plays an insignificant role in this signaling pathway. Therefore, OsCSN2 affects the growth and development of rice seedlings through the regulation of OsSLR1 in the GA signaling pathway. In this process, OsSLR1 acts as a negative regulator. The downregulation of OsSLR1 expression can promote the growth and development of rice seedlings, which is reflected in their growth and development. It is reflected in the elongation of the stems, the coleoptile, and the first incomplete leaf of the seedlings, with less effect on the roots.

Under blue light, the mRNA levels of *OsCOP1*, *OsCRY2*, *OsCUL4*, and *OsBBX14* in *OsCSN2-OE* were relatively high and tended to increase (Figure 5B). Previous studies have demonstrated that these genes play a regulatory role in blue light morphogenesis in rice [17,25]. The relative expression levels of *oscsn2-1* and *oscsn2-2* both tended to be low (Figure 5B). After synergic treatment with blue light and the exogenous hormone GA_3_, the relative expression levels of *OsCOP1* and *OsBBX14* in *OsCSN2-OE* plants tended to decrease, whereas those of *OsCUL4* tended to increase, and the addition of the exogenous hormone *OsCSN2K67E-OE* resulted in significant changes (Figure 5C). The overall *OsCOP1*, *OsCRY2*, and *OsBBX14* genes showed different upward trends after the addition of PAC (Figure 5D). Some studies have demonstrated that OsBBX14 is involved mainly in blue light-regulated photomorphogenesis; OsBBX14 is involved in the OsCRY-mediated light pathway and interacts with OsCRY2, resulting in the specific dwarf phenotype of *OsBBX14*-overexpressing mutants under blue light [25]. The COP9 signaling complex regulates COP1. The required recombinant vectors were constructed, and the plasmid were cotransformed into yeast strain AH109 for yeast two-hybrid assay to verify the interactions between OsCSN2 and OsCUL4, which were cultured in SD-Trp-Leu media versus SD-Trp-Leu-His-Ade media. All positive controls and negative controls could grow normally on SD-Trp-Leu media, indicating that there was no problem with the experimental materials. On SD-Trp-Leu-His-Ade media, the cotransformed strains of the positive control pGADT7 + pGBKT7-53 and the experimental group pGADT7-OsCSN2 + pGBKT7-OsCUL4 grew normally, whereas the negative pGADT7 + pGBKT7-Lam, pGBKT7 + pGADT7-OsCSN2, and pGADT7 + pGBKT7-OsCUL4 cotransformed strains were unable to grow (Figure 5E). These results indicated that OsCSN2 and OsCUL4 could interact. OsCSN2 may regulate the growth and development of rice seedlings through OsCUL4 in Cullin-RING E3. The results of the BIFC experiment revealed yellow fluorescence when pSAT4A-nEYFP-OsCSN2 and pSAT4A-cEYFP-OsCUL4 coinfected the inner epidermis of the onion (Figure 5F), which also supported the finding that OsCSN2 and OsCUL4 could interact.

In summary, the expression level of *OsBBX14* in *OsCSN2-OE* was significantly greater than that in the WT and knockout mutant lines, suggesting the inhibitory role of OsBBX14 in blue light morphogenesis [25]. The changes in phenotype characteristics and gene expression levels confirmed that OsCSN2 may be a negative regulator under blue light conditions and acts synergistically with OsBBX14. OsCSN2 may coordinate the degradation of SLR1 in the GA signaling pathway through the blue light receptors CRY2 and COP1 to regulate rice seedlings. OsCSN2 also regulates this process through the CUL4-based E3 ubiquitin ligase (Figure 6). Ultimately, endogenous hormone changes and phenotypic growth and developmental changes were most significant in rice seedlings aboveground. 

## 3. Discussion

### 3.1. OsCSN2 as a Potential Negative Regulator Under Blue Light

The COP9 signaling complex is a large multi-subunit protein complex composed of eight subunits, and this structural composition is highly conserved in higher eukaryotes and has been identified as an essential factor of photomorphogenesis in *Arabidopsis*, a negative regulator of the blue light receptor, which acts downstream of the blue light receptor [28,29]. A change in any subunit in the COP9 signaling complex triggers the integrity and function of the complex, resulting in developmental problems in the organism [30]. However, few studies have investigated the COP9 signaling complex in rice. We treated the WT and the *OsCSN2* mutants with blue light, and after the addition of the exogenous hormones GA_3_ and PAC, we observed different behaviors in the rice seedlings.

Under natural light treatment, the *OsCSN2* knockout mutants and weak expression mutants presented greater growth trends, whereas the *OsCSN2* overexpression mutants presented a shorter phenotype (Figure 1A). Therefore, it was speculated that CSN2 plays a role in the response of rice to light as a negative regulator of morphogenesis. The growth inhibition was present in all parts of the WT and mutant under blue light, especially in the aboveground part of rice seedlings, where the WT was higher than the mutant lines, and the *OsCSN2* knockout mutant grew better than the overexpression mutant (Figure 1A). Therefore, OsCSN2 may be a negative regulator of blue light-regulated growth in rice seedlings. Blue light inhibits the synthesis of endogenous gibberellin. *OsCSN2* mutants showed particularly sensitive aboveground expression with phenotypic restoration or promotion following the addition of the exogenous hormone GA_3_ under blue light, but the changes in the roots were not very significant (Figure 3A). After different ubiquitination loci changed, the *OsCSN2K67E-OE* mutants presented the most obvious regulation of hormones under blue light and hormone co-treatment, which may be related to the ubiquitination locus. OsCSN2 may rescue the inhibitory effect of blue light on endogenous hormones by exogenous hormone addition and promote the aboveground growth and development of rice. However, *OsCSN2* in rice seedling roots is not regulated through the GA signaling pathway under blue light. In summary, OsCSN2 may play a negative regulatory role under blue light.

### 3.2. In Rice, the Subunits of the COP9 Signaling Complex in the GA Signaling Pathway Exhibit Different Functions Under Blue Light

In the COP9 signaling complex, CSN1 and CSN2 are two essential subunits. Previous experiments have shown that the effect of *OsCSN1-OE* under natural light is significantly lower than that of the WT, and the levels of the knockout mutants and weak expression mutants are significantly greater than those of the WT [17]. Specifically, OsCSN1 negatively regulates the growth and development of rice seedlings, and OsCSN1 also negatively regulates the growth of rice seedlings under blue light [17]. This result was consistent with that of OsCSN2. Therefore, it was speculated that OsCSN1 and OsCSN2 subunits are also negative regulators of the light signaling pathway, but there are large differences in the expression levels of related genes in rice. Rice plant height also has an essential effect on rice yield. Maintaining rice plant height at a stable optimal height is the key to maximizing rice yield. Gibberellin has a certain regulatory effect on rice stem elongation and can promote stem elongation [31]. In the GA signaling pathway, the DELLA protein is a key protein that regulates this pathway, and there are five members in *Arabidopsis* [26]. The DELLA protein in rice is SLR1, which negatively regulates the GA signaling pathway [27]. Under natural light, the OsSLR1 gene and protein expression levels were greater in the *OsCSN2-OE* mutants (Figure 4A and Figure 5A), and all the rice seedlings tended to be shorter (Figure 1A). After the addition of exogenous GA_3_ under blue light, the height of the *OsCSN2* mutants was restored compared with that under natural light (Figure 1A). More notably, the growth of the first incomplete leaf and coleoptile of the *OsCSN2* mutant was significantly promoted by the addition of the exogenous hormone GA_3_ under blue light compared with natural light (Figure 1A), whereas the aboveground part of the *OsCSN1* mutant was not restored to the natural state and remained in a suppressed state [17]. Therefore, OsCSN2 can restore and promote aboveground growth in response to the GA signaling pathway under blue light, rescuing the aboveground growth and development of the *OsCSN2* mutant. However, the changes in the roots were not obvious. OsCSN2 may be involved in the GA signaling pathway to affect the growth and development of the root system. Like that of OsCSN1, the hormone signaling pathway by which OsCSN2 effectively regulates root growth remains to be studied. These results suggest that OsCSN1 and OsCSN2 have different functions in rice. CSN1 and CSN2 likely play different functions in the GA signaling pathway under blue light by regulating the COP9 signaling complex.

The loss of the *CSN2* gene function does not result in the complete loss of the functions of the other subunits of the COP9 signaling complex, but it affects the relative expression levels of its genes. The CSN complex shows two symmetric modules, with CSN1/2/3/8 as one of the blocks. In this module, CSN2 is connected only to CSN1, and the two modules are connected to CSN6 through CSN1 [32]. The deletion or overexpression of the *CSN2* gene in rice may significantly change the overall structure of the COP9 signaling complex, resulting in functional changes and unknown changes in other subunits. Under blue light, the expression level of OsCSN1 in *OsCSN2-OE* increased (Figure 4B and Figure 5B). After the addition of the exogenous hormone GA_3_, the overall relative expression level of OsCSN1 was more significantly inhibited, but the expression level of *OsCSN2-OE* was still relatively high (Figure 4C and Figure 5C). Coincidentally, under blue light, the expression level of the OsCSN2 protein and the relative expression level of the gene in *OsCSN1-OE* significantly increased [17]. It is possible that both *CSN1* and *CSN2* play negative regulatory roles and that the two genes are directly connected and interact. Under blue light, CSN2 may affect the assembly of the COP9 signaling complex through the GA signaling pathway, thereby affecting the formation of CSN1 monomeric subunits. CSN1 inhibits the growth of rice seedlings under blue light [17], and CSN2 is likely to play a synergistic role with CSN1. Therefore, the COP9 signaling complex may play a negative regulatory role in the growth of rice seedlings, and the decrease in the CSN1 protein expression through the exogenous addition of the GA_3_ hormone is likely to inhibit the formation of the COP9 signaling complex, thereby promoting the growth and development of rice seedlings. Under natural light and blue light, the expression level of OsCSN5 in the *OsCSN2* mutants relatively decreased in the overexpression mutants, whereas the knockout and weak expression mutants presented relatively high expression levels (Figure 4A,B). There was a decreasing trend compared with the WT level. We speculate that this is related to the structure of the CSN. CSN5 has a metal domain and can also exist and function independently [33]; this affects the synthesis of the CSN. However, after the ubiquitination site changed, the expression level of OsCSN5 in *OsCSN2K67E-OE* tended to increase under blue light (Figure 4B). The deubiquitination of ubiquitination site 67 strongly affected the formation of OsCSN5; it has a promoting effect. However, after the addition of the exogenous hormone GA_3_, the gene expression of *OsCSN2K67E-OE* decreased (Figure 4C). Therefore, the specific molecular mechanism remains to be further studied.

### 3.3. OsCSN2 Regulates the GA Signaling Pathway Through a CUL4-Based E3 Ubiquitin Ligase

The COP9 signaling complex controls cellular function through multiple regulatory roles with Cullin-RING E3 ubiquitin ligases and the degradation of key signaling proteins, and binds to and functions with protein kinases and deubiquitylating enzymes (DUBs) to regulate CSN, CRLs, and CRLs substrates [34]. CSN regulates the phosphorylation of ubiquitin–proteasome pathway substrates through related kinases, and CRLs perform deubiquitination to regulate the activity of Cullin-RING E3 ligases and affect intracellular protein degradation pathways [35]. The gibberellin signaling pathway is a key regulator of plant growth. An essential component of this pathway is the DELLA protein, which binds to gibberellin to regulate plant growth and development and promotes stem elongation.

Under blue light, the content of the endogenous hormone GA_3_ was significantly inhibited, and the content of *OsCSN2-OE* was relatively high (Figure 2A–C). After the addition of the exogenous hormone GA_3_, the content of the endogenous hormone GA_3_ in *OsCSN2-OE* was most inhibited, while the content of *OsCSN2K67E-OE* was significantly increased, which may be related to the ubiquitination sites (Figure 2A–C). The results of yeast two-hybrid and BIFC experiments revealed that there was an interaction between OsCSN2 and OsCUL4 (Figure 5E,F). Under blue light, the relative expression level of the *OsCUL4* gene in *OsCSN2-OE* was upregulated (Figure 5B). After the addition of the exogenous hormone GA_3_, the relative expression level of the *OsCUL4* gene in *OsCSN2-OE* and *OsCSN2K67E-OE* was significantly increased, whereas the difference between the *OsCSN2* knockout and weak expression mutants was not significant (Figure 5C). It was speculated that OsCSN2 regulated CUL4 in the Cullin-RING protein family through the assembly of the COP9 signaling complex, thereby mediating the expression of the related gene *SLR1* in the GA signaling pathway. Therefore, the synthesis of endogenous hormones in rice seedlings and the growth and development of the aboveground parts of rice seedlings are affected. Owing to the change in amino acid 67 of OsCSN2, it is sensitive to blue light and the expression of GA signaling pathways in related genes and proteins, and the effect of deubiquitination on OsCSN2 is greater. Therefore, OsCSN2 synergizes with the exogenous hormone GA_3_ in blue light to promote the growth and development of aboveground parts of rice seedlings, especially in the coleoptile and the first incomplete leaf. In contrast, OsCSN1 does not have this effect under the synergistic effect of OsCSN1 [17].

In the presence of blue light and the addition of the exogenous hormone GA_3_, the *OsCOP1*, *OsCRY2*, and *OsBBX14* genes of *OsCSN2-OE* all presented relatively high expression levels at the mRNA level, whereas the relative expression levels of the knockout mutants were low (Figure 5C). OsBBX14 has been confirmed to be involved mainly in blue light-regulated photomorphogenesis [25]. During the growth process of rice seedlings, they sense blue light signals through blue light receptors. The overexpression of OsCRY2 under blue light can significantly shorten the total length of seedlings and the lengths of the main leaves and coleoptiles [36]. OsCSN2 regulates the growth and development of rice plants under blue light through the GA signaling pathway. There are many related genes and complex and complicated regulatory mechanisms involved in this process, and it is speculated that OsBBX14 negatively regulates the growth and development of CSN2 rice seedlings under blue light. OsCRY2, as a receptor of the sensing blue light signaling pathway, functioned in interaction with OsCOP1, resulting in the suppression of rice seedling plant height and the elongation of the first incomplete leaf, and also verified that the COP9 signaling complex may regulate COP1 to affect blue light morphogenesis. However, the specific molecular mechanism is still not clear and will continue to be studied in the future.

## 4. Materials and Methods

### 4.1. Plant Material

Wild-type rice Nipponbare (WT), *OsCSN2* deletion mutant (*oscsn2-1*), *OsCSN2* weak expression mutant (*oscsn2-2*), *OsCSN2* overexpression mutant (*OsCSN2-OE*), and *OsCSN2* point mutants (*OsCSN2K64E-OE*, *OsCSN2K67E-OE*, and *OsCSN2K104E-OE*) were obtained from our laboratory [27].

### 4.2. Experimental Methods

#### 4.2.1. Rice Seedling Culture Conditions and Environment

The husks of the wild-type Nipponbare and rice mutant plants were removed in advance, soaked in 75% ethanol, disinfected for 10 min, washed once with sterile water, disinfected with 70% NaClO for 30 min, and finally rinsed with sterile water for approximately 6 min. The seeds of the wild-type Nipponbare variety and rice mutants were evenly divided into 4 groups. Agarose media at a concentration of 0.8% (*w*/*v*) were added (the 2 groups were separately supplemented with gibberellin (GA_3_) at a concentration of 10mM and paclobutrazol (PAC) at a concentration of 100 mM). The control group was placed in a light incubator (203 µmol/m^2^/s light, a temperature of 28 ± 1 °C, and a culture cycle of 12 h of light and 12 h of darkness), and the other three groups were placed in a blue light incubator (30 µmol/m^2^/s light, the temperature was 28 ± 1 °C, and the blue light source was LED-B, which was continuously illuminated with blue light).

#### 4.2.2. Analysis of Rice Seedling Phenotypes and Endogenous Hormones

After 9 days of culture, the plant height, root length, coleoptile, and first incomplete leaf height of the rice seedlings were measured and analyzed. First, 1g of the aboveground parts of 9-day-old rice seedlings were weighed for liquid nitrogen milling, and the extraction method was referred to Ye’s method [37]. The aboveground rice samples, which had been fully ground by liquid nitrogen, were immediately added with 80% cold methanol, mixed thoroughly, and then centrifuged at 4 °C for 15 min at 5000× *g*. The supernatant was aspirated and filtered through a Sep-PaK PlusC18 column. The filtrate was dried in a vacuum oven and then dissolved in methanol for use in subsequent experiments. Enzyme-linked immunosorbent assay (ELISA) was used to determine the OD value at 450 nm for the determination and calculation of endogenous hormone GA_3_ content in rice seedlings. The ELISA kit used was purchased from Jiangsu Jingmei Biotechnology Co. (Yancheng, China). The experiment was repeated 3 times for each group of samples (N = 3). For the experiments, Excel 2019 and IBM SPSS Statistics 26 (SPSS, https://www.ibm.com/cn-zh/products/spss statistics, accessed on 14 February 2024) were used to analyze the data, and GraphPad Prism 8 (https://www.graphpad-prism.cn/) was used to draw the chart. The data are expressed as the means ± standard errors of the control group and the experimental group. Statistical significance was set at *p* ≤ 0.05.

#### 4.2.3. Protein Extraction, Western Blot Experiments/Antibodies, and Western Blot Analysis of Rice Seedlings

Protein was extracted after 0.5 g of seedlings were ground in liquid nitrogen. The formula used for protein extraction was RIPA lysis buffer (Genstar, Beijing, China), 1 mM DTT, 1 mM PMSF, and 1 × protease inhibitor cocktail (Roche, China). The mixture was centrifuged at 15,000 rpm at 4 °C for 10 min, the supernatant was transferred to a new centrifuge tube, 5 × SDS (Genstar, China) was added, the mixture was mixed thoroughly, and the mixture was boiled for 10 min to denature the tube [17]. The protein samples were added to 10% SDS–PAGE gels for subsequent Western blot (WB) experiments. All the antibodies used in this study were from Wuhan ABclonal Biotechnology Co. (ABclonal, Wuhan, China). The primary antibodies used were anti-OsCSN1, anti-OsCSN5, anti-OsSLR1, and anti-OsABI5. To visualize the signals, the Universal Hood III system (731BR03292, BIO-RAD) was used.

#### 4.2.4. Total RNA Extraction and Quantitative Real-Time Polymerase Chain Reaction

Total RNA was extracted from seedling material using the Spectrum Plant Total RNA Kit (Sigma-Aldrich, Steinheim, Germany), and RNA was reverse-transcribed into cDNA using StarScript II versus starting material for cDNA synthesis using a gDNA remover (Genstar, China). Quantitative real-time PCR (qRT–PCR) experiments were subsequently performed. The experiment was completed on a StepOne RT–PCR Plus™ Real-Time PCR System (Applied Biosystems, Thermo Fisher Scientific, Beijing, China) using ROX in 2× RealStar Green Fast Mix (GenStar, China). The blank control gene GAPDH (JN848809) was used to calculate the relative mRNA levels of the three averages of the experiment. Table 1 lists the detailed information of the gene-specific primers used for qPCR and RT–PCR (Table 1).

#### 4.2.5. Yeast Two-Hybrid Experiment

The yeast two-hybrid experiment was performed according to Zhang’s method [38]. The pGADT7 and pGBKT7 plasmids were purchased from Miaoling Plasmid Platform (Wuhan, China). The experimental groups pGADT7-OsCSN2 + pGBKT7-OsCUL4, the positive control pGADT7 + pGBKT7-53, and the negative controls pGADT7 + pGBKT7-Lam, pGBKT7 + pGADT7-OsCSN2, and pGADT7 + pGBKT7-OsCUL4 were co-transformed into the yeast strain AH109 for yeast two-hybrid analysis.

#### 4.2.6. Bimolecular Fluorescence Complementation Experiment

Bimolecular fluorescence complementation experiments are abbreviated as BIFC. The pSAT4A-cEYFP-N1 and pSAT4A-nEYFP-N1 plasmids were purchased from Huizao Biotechnology Co. (Wuhan, China). The negative controls pSAT4A-cEYFP and pSAT4A-nEYFP, pSAT4A-cEYFP and pSAT4A-nEYFP-OsCSN2, pSAT4A-nEYFP and pSAT4A-cEYFP-OsCUL4, and the experimental groups pSAT4A-nEYFP-OsCSN2 and pSAT4A-cEYFP-OsCUL4 were constructed and co-transformed into agricultural products. The Bacillus strain GV3101 was used to infect the inner epidermis of onion and was subsequently subjected to bimolecular fluorescence complementation experiments. The observation instrument used was a confocal laser microscope (TCS SP8) from Leica AG (Wetzlar, Germany).

## 5. Conclusions

Our study revealed that OsCSN2 may be a crucial negative regulator of blue-light-mediated photomorphogenesis in rice seedlings. By modulating the GA signaling pathway, OsCSN2 influences the growth and development of aboveground plant parts in response to blue light. Specifically, OsCSN2 perceives blue light signals through CRY2, regulating the expression of COP1 and BBX14, which in turn affects the degradation of SLR1 and the subsequent elongation of plant height, coleoptiles, and the first incomplete leaf. This study not only underscores the significant role of OsCSN2 in blue light signaling but also highlights the distinct functions of OsCSN1 and OsCSN2 within the COP9 signaling complex in rice photomorphogenesis. These findings provide valuable insights into the molecular mechanisms underlying plant responses to light, potentially informing strategies for optimizing growth conditions in agricultural settings.

## Figures and Tables

**Figure 1 plants-14-02015-f001:**
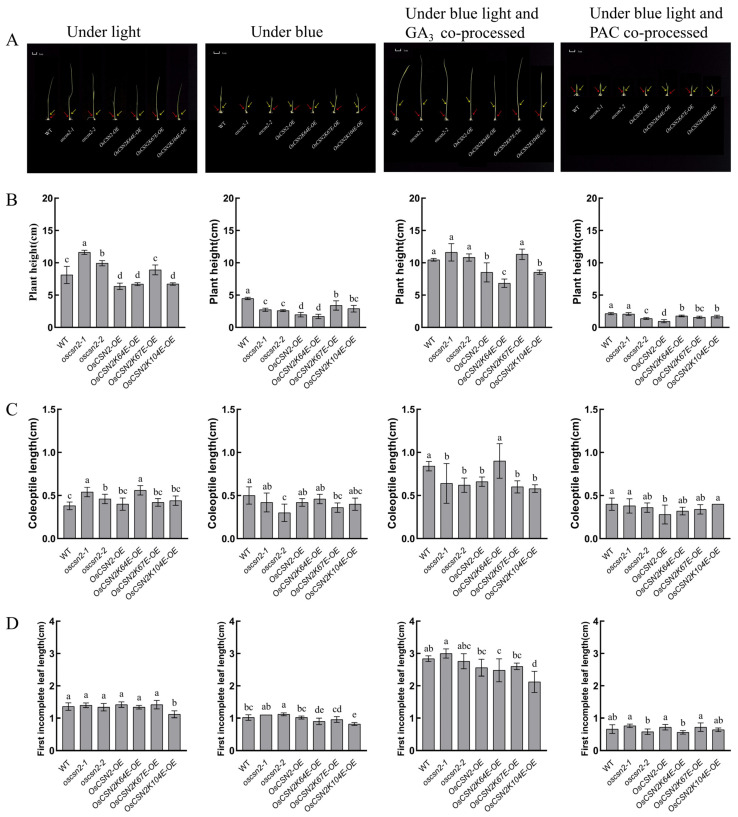
Plant height, coleoptile length, and first incomplete leaf of WT and *OsCSN2* mutants under different condition treatments. The red arrow in the figure points to the coleoptile, and the yellow arrow in the figure points to the first incomplete leaf. (Different letters within the same group indicate significant differences (*p* ≤ 0.05), while the same letters indicate no significant differences.) (**A**) The aboveground of WT and *OsCSN2* mutants grown for 9 days under natural light, blue light, blue light and GA_3_ co-treatment, and blue light and PAC co-treatment. (**B**) Plant height data graph of WT and *OsCSN2* mutants grown 9 days under light, blue light, blue light and GA_3_ co-processed, blue and PAC co-processed. (**C**) Coleoptile length data graph of WT and *OsCSN2* mutants grown 9 days under light, blue light, blue light and GA_3_ co-processed, and blue light and PAC co-processed. (**D**) First incomplete leaf length data graph of WT and *OsCSN2* mutants grown 9 days under light, blue light, blue light and GA_3_ co-processed, and blue light and PAC co-processed.

**Figure 2 plants-14-02015-f002:**
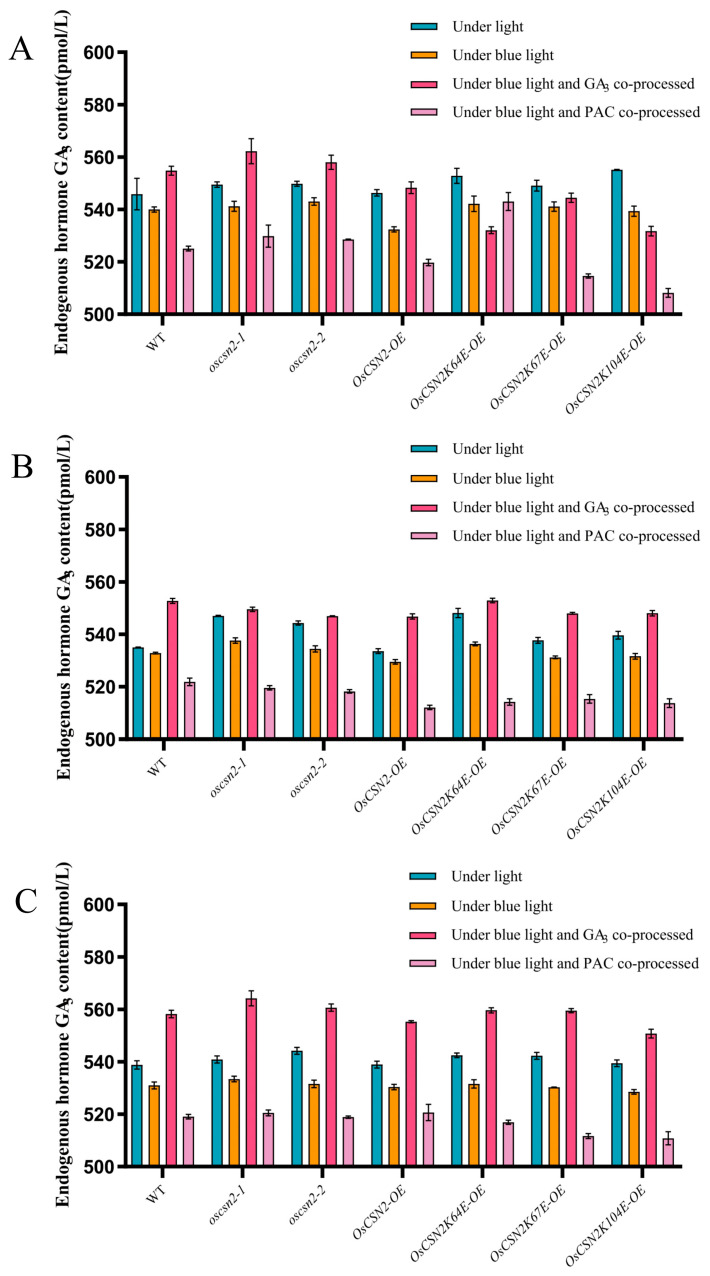
Endogenous hormone GA_3_ content measured at different sites of WT and *OsCSN2* mutants grown under different conditions for 9 days. (**A**) Data plots of the content of the endogenous hormone GA_3_ in stems of WT and *OsCSN2* mutants grown for 9 days under light, blue light, co-treatment of blue light and GA_3_, and co-treatment of blue and PAC. (**B**) Data plots of the content of the endogenous hormone GA_3_ in coleoptile length of WT and *OsCSN2* mutants grown for 9 days under light, blue light, co-treatment of blue light and GA_3_, and co-treatment of blue and PAC. (**C**) Data plots of the content of the endogenous hormone GA_3_ in first incomplete leaf of WT and *OsCSN2* mutants grown for 9 days under light, blue light, co-treatment of blue light and GA_3_, and co-treatment of blue and PAC.

**Figure 3 plants-14-02015-f003:**
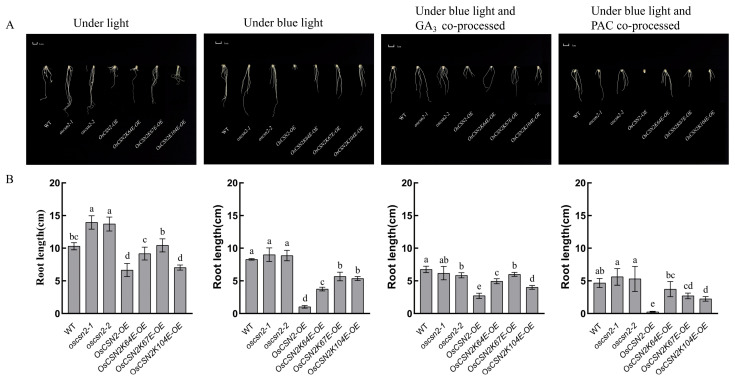
Root length of WT and *OsCSN2* mutants under different condition treatments. (**A**) Root length of WT and *OsCSN2* mutants grown for 9 days under natural light, blue light, blue light and GA_3_ co-treatment, and blue and PAC co-treatment. (Different letters within the same group indicate significant differences (*p* ≤ 0.05), while the same letters indicate no significant differences.) (**B**) Root length data graph of WT and *OsCSN2* mutants grown 9 days under light, blue light, blue light and GA_3_ co-processed, and blue and PAC co-processed.

**Figure 4 plants-14-02015-f004:**
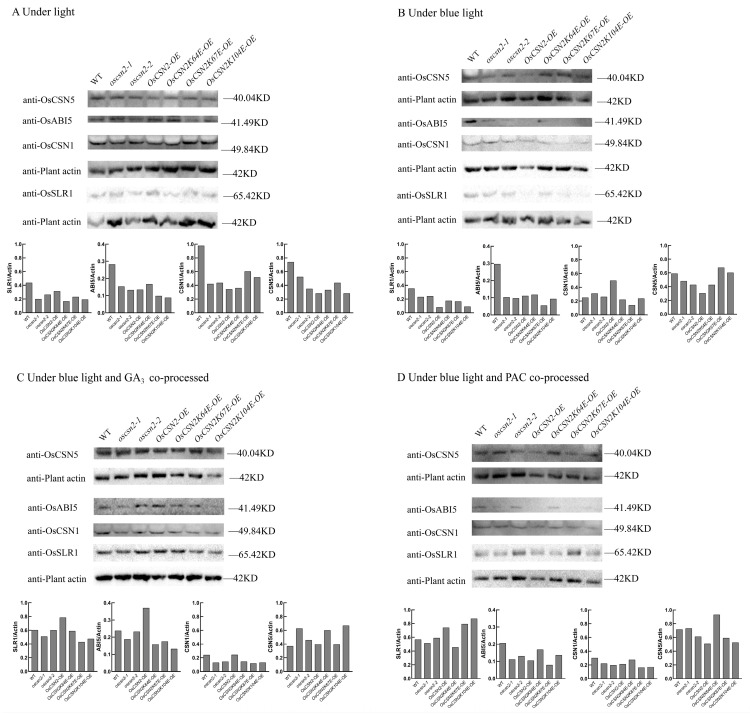
Changes in expression of related proteins under different treatments in WT and *OsCSN2* mutants. (**A**) Under light. (**B**) Under blue light. (**C**) Under blue light and GA_3_. (**D**) Under blue light and PAC.

**Figure 5 plants-14-02015-f005:**
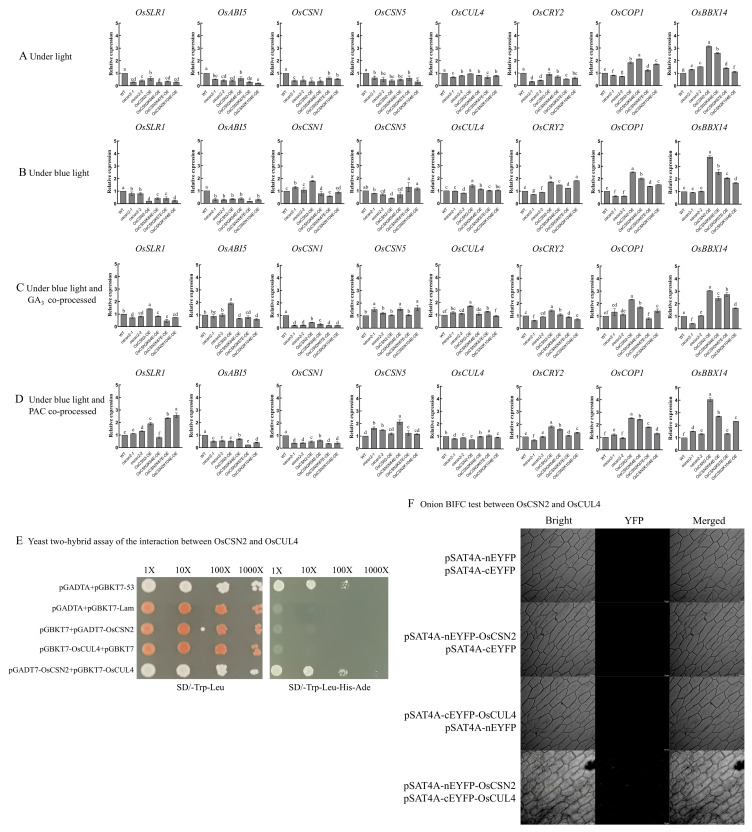
Experimental map of changes in OsCSN2-related genes and protein interactions during the growth of rice seedlings under blue light. Changes in the expression of genes associated with mRNA levels in WT and *OsCSN2* mutants after 9 days of growth under different treatment conditions. Expression of *OsSLR1*, *OsABI5*, *OsCSN1*, *OsCSN5*, *OsCUL4*, *OsCRY2*, *OsCOP1*, and *OsBBX14* genes in the sample lines was examined using qPCR. (Different letters within the same group indicate significant differences (*p* ≤ 0.05), while the same letters indicate no significant differences.)

**Figure 6 plants-14-02015-f006:**
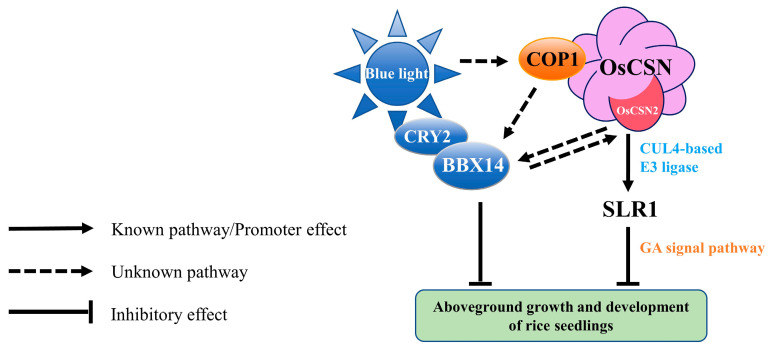
Putative signaling pathway for blue light-mediated regulation of rice seedling growth and devel-opment by OsCSN2.

**Table 1 plants-14-02015-t001:** All the primers that were used in this study.

Primer	Sequence (5′-3′)	Purpose
Dye-SLR1F	CATGCTTTCCGAGCTCAACG	q RT-PCR
Dye-SLR1R	TGACAGTGGACGAGGTGGAA	q RT-PCR
Dye-CSN1F	CGGCCCGTAAGTTTGTTGAG	q RT-PCR
Dye-CSN1R	AGGGCACCATAGACAGCAAC	q RT-PCR
Dye-CSN5F	GAGCAAGCTGAGGGTCAACT	q RT-PCR
Dye-CSN5R	GACCATGGACCTHTTCAGCA	q RT-PCR
Dye-ABI5F	TGGTAGACAGTGGAAGCGGAAG	q RT-PCR
Dye-ABI5R	ACAGCGGTAGCGGCAAGG	q RT-PCR
Dye-CUL4F	AGGACAGACAGTATCAGGTGGATGC	q RT-PCR
Dye-CUL4R	TCCGATGGCTTGATTGGGAACTTG	q RT-PCR
Dye-CRY2F	TGCTGATCCGAGCCGAGAGTAC	q RT-PCR
Dye-CRY2R	ACGCACAAGAGAAACAGGGTCATAC	q RT-PCR
Dye-COP1F	CATCTCAGCCACAAGAGCGACTG	q RT-PCR
Dye-COP1R	GGTCTATCGGTGATGCTGTCTTCG	q RT-PCR
Dye-BBX14F	GCCGCCGACCAAGAGGAG	q RT-PCR
Dye-BBX14R	CCGAAGCCGAAGCCAAAGC	q RT-PCR

## Data Availability

The data presented in this study are available in the article.

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
