# Peer review of "Under Blue Light Treatment, OsCSN2 Regulates the Phenotype of Rice Seedlings Through the GA Signaling Pathway"

_plants, 2025, doi:10.3390/plants14132015_

Round 1
Reviewer 1 Report
Comments and Suggestions for Authors
This is an interesting study investigating how OsCSN2, a subunit of the COP9, influences rice seedling growth under blue light via the GA treatments. The authors utilise genetic mutants, hormone treatments, and molecular biology techniques (western blot, two-hybrid and BiFC) to uncover OsCSN2's regulatory role in photomorphogenesis. I have a couple of suggestions to help improve clarity and structure, which I hope the authors will find constructive:
Several acronyms and gene/protein names are introduced without a full definition at first mention. For example:
- Line 2 (Abstract) and Line 39 (introduction): COP9 is used before introducing Constitutive Photomorphogenesis protein in Line 65.
- Line 45–46: The gene COPS2 is mentioned alongside several other names (Sgn2, TRIP15, hAlien, FUS12) without sufficient context.
- Line 61: CRL
- Line 63: SCF
- Line 93: ZTL
- Line 97: HY5
- Among many others in the manuscript.
I recommend carefully reviewing the manuscript to ensure that each acronym or genes/proteins name is spelt out in full when first introduced.
Check consistency in gene/protein/mutant names across the manuscript. For instance, line 155 “oscsn2-1 and oscsn2-2” is not in italics.
The transition from paragraph two (focused on COPs, photomorphogenesis, and ubiquitin ligases) to paragraph three (focused on the importance of rice in China) feels abrupt. Consider adding a bridging sentence to improve the logical connection between paragraphs.
The sentence in lines 79–81 ("...chlorophyll in hemp...") could benefit from broader evidence across plant species. If similar findings exist in other plants or crops, add them to enhance the general value of the statement.
The short sentence "CRY was first found in Arabidopsis [14]" (Line 84) does not seem to add significant value. If there is a deeper point to be made, such as its relevance or conservation across species, please consider expanding or clarifying the statement.
The meaning of the following sentence is unclear and could be confusing: "However, the height of AtHY5-overexpressing Arabidopsis plants was the same as that of wild-type plants. The plant height was lower than that of the wild type [23]." It appears contradictory.
In the paragraph beginning at line 117, the discussion starts with phytohormones and shifts quickly to OsCSN2 mutants without a clear connection. Additionally, some sentences in this part of the introduction read more like results or methods than background or context.
In general, I would recommend refining the structure and logic flow in the manuscript.
Figures 1A and 2A are identical, with the only difference being the presence of arrows. Additionally, Figure 4 is introduced very early in the manuscript and appears second in the text. I suggest combining figures 1, 2, and 4 into a single comprehensive figure, if possible. Provide statistical analysis for the graphs in figure 4.
In the results section, the headlines are often framed as strong causal statements, suggesting definitive regulatory roles for OsCSN2 (e.g., “OsCSN2 Can Negatively Regulate the Inhibitory Effect of Blue Light…Through the GA Signaling Pathway” or “COP1 and BBX14 Regulate OsCSN2…via SLR1 Degradation”). However, based on the data presented, these conclusions appear to rely primarily on phenotypic and correlative observations, without sufficient molecular evidence to support direct causality. I recommend rephrasing the section titles to more accurately reflect the nature and strength of the evidence, in order to avoid overinterpretation and better align the manuscript narrative with the experimental findings.
Likewise, in the Discussion and Conclusion sections, some statements appear to overinterpret the data. For instance, the sentence “Our study revealed that OsCSN2 is a crucial negative regulator…” would be more appropriately phrased as “Our study provides evidence supporting a role for OsCSN2 as a negative regulator…” This more accurately reflects the real strength of the evidence and avoids not proven definitive causality.
While the data support an association between OsCSN2 and SLR1 degradation, the mechanistic link remains largely correlative. Direct evidence demonstrating that OsCSN2 modulates SLR1 protein stability would significantly strengthen this claim. Additionally, the study would benefit from broader contextualisation of blue light and GA crosstalk in rice or other monocots. Finally, the functional significance of the OsCSN2 point mutations (K64E, K67E, K104E) is intriguing, but the manuscript could better integrate those findings into the mechanistic narrative, particularly how these mutations might influence CSN2’s role within the COP9 complex or its interactions with CUL4.
Comments on the Quality of English LanguageOverall, the manuscript requires improvements in the redaction and readability.
Reviewer 2 Report
Comments and Suggestions for Authors
Article Review by Xinhai Yu : Under Blue Light Treatment, OsCSN2 Regulates the Phenotype of Rice Seedlings Through the GA Signaling Pathway.
The work examines the role of the COP9 signaling complex (CSN) in rice under blue lighting conditions. The complex consists of several subunits and plays a key role in the regulation of plant photomorphogenesis. Using OsCSN2 (KO) knockout mutants and overexpression (OE), suppression of the general phenotype of seedlings was shown when exposed to blue light, indicating that OsCSN2 acts as a negative regulator of blue-light-mediated morphogenesis. It was shown that exogenous administration of gibberellin (GA3) and the GA synthesis inhibitor paclobutrazole (PAC) modulated the elongation of seedlings in response to exposure to blue light. Real-time PCR (qRT-PCR) has shown that OsCSN2 is an important sensor of blue light signals, influencing COP1 and BBX14 expression and interacting with cryptochrome (CRY2), which highlights its role in the photoreceptive signaling pathway. The results also highlight the different roles of OsCSN1 and OsCSN2 in CSN in modulating rice seedling photomorphogenesis under the action of blue light.
In the methodology, the authors write: The control group was placed in a light incubator (203 μmol/m 2/s light, the temperature of 28 ± 1 °C, a culture cycle of 12 h of light, and 12 h of darkness), the other three groups were placed in a blue light incubator (30 μmol/m 2/s light, the temperature was 28 ± 1 °C, and the blue light source was LED-B, which was continuously illuminated with blue light). It should be noted that interaction with cryptochrome implies a sharp change in the effect of blue light on the signal complex depending on the radiation spectrum, as well as the "shutdown" of cryptochrome in long-wavelength blue light. (A. Y. Khudyakova, A. A. Kosobryukhov, P. P. Pashkovsky, and V. D. Kreslavski, Cryptochromes and Their Role in the Process of Plant Adaptation Russian Journal of Plant Physiology, 2024, Vol. 71:42.) In this regard, the methodology should specify under what (specific) conditions plants were irradiated in an experiment.
The work is, without a doubt, a further contribution to the study of the mechanism of light signaling and deserves publication in the journal.
Author Response
Thank you for your recognition and affirmation of this manuscript – we are extremely honored. We would also like to express our sincere gratitude for your diligent reading of this article. We wish you a pleasant life and every success in your work. Best wishes to you!
Reviewer 3 Report
Comments and Suggestions for Authors
The manuscript under review is devoted to the study of the pathway of light signal transmission during blue light irradiation of rice plants at the stage of early seedling development during the transition from heterotrophic to autotrophic growth under the influence of light. The main issue considered by the authors in this study is related to the identification of the role of individual subunits of the COP9 signaling complex, namely, its key subunit CSN2, which is involved in the assembly and functioning of this complex. An undoubted advantage of this study is the choice of the object, since rice is an important agricultural crop used to feed a large part of the world's population. In this regard, the search for ways to effectively use solar energy by plants and, in particular, economically important species, seems to be an extremely urgent task. The authors used rice seedlings as objects of study, representing mutations in the OsCSN2 gene, characterized by different levels of its expression, obtained in the laboratory earlier. By exposing seedlings to blue light, the authors demonstrated the role of OsCSN2 as a negative regulator of morphogenesis, which can be modulated by exogenous exposure of seedlings to gibberellin (GA3) or an inhibitor of its synthesis. The studies showed the participation of OsCSN2 in the inhibitory effect of blue light on the development of aboveground organs of rice plants through the gibberellin signaling pathway, as well as the differential role of OsCSN1 and OsCSN2 in CSN in modulating photomorphogenesis of rice seedlings. The idea of ​​​​the complex regulatory mechanisms governing the plant response to blue light is summarized by the authors in the form of a graphical figure reflecting the role of OsCSN2 in the putative signaling pathway for rice plants. The authors mention that the first work in this direction was carried out on Arabidopsis thaliana plants and the cited work dates back to 2020. It would be interesting to compare both of these models and add additional information about A.thaliana to Figure 7. The methods and approaches used by the authors are generally accepted and do not require any additional improvements in terms of methodology. The data obtained by the authors are presented in the figures quite clearly, but the authors should pay attention to the violation of the order of presentation of the figures - after figure 1 there is figure 4. All references given in the list of cited works are appropriate and correspond to the field of research in which this work was carried out. The manuscript can be recommended for publication in the journal in the presented form, taking into account the minor comments mentioned above.
Author Response
Thank you for reviewing and affirming this manuscript. Based on your valuable comments, I have made revisions to it.
Comments 1: Authors should pay attention to the violation of the order of presentation of the figures - after figure 1 there is figure 4.
Response 1: Thank you for your comments. I have revised the figures, as specified in the text where the figures are located.
Round 2
Reviewer 1 Report
Comments and Suggestions for Authors
First, I would like to thank the authors for addressing the previous comments. I have no further major concerns. Below are a few last minor suggestions:
Line 19: Consider changing “Utilizing OsCSN2 knockout (KO) and overexpression (OE) mutants…” to “Utilizing OsCSN2 knockout (KO) mutant plants and transgenic overexpression (OE) lines for wild-type (WT) and mutated versions of OsCSN2…” This clarification helps avoid confusion, as the original phrasing could imply that the overexpression lines result from a mutation (e.g., in the promoter), rather than being transgenic lines expressing either the WT or a mutated version of the coding sequence. According to reference 27, from which these lines are derived, the OE lines are indeed transgenic lines carrying either the WT or mutated OsCSN2 CDS. This distinction is especially important in the abstract or introduction. Later in the manuscript, once the data and figures with the mutated versions for OSCSN2 are presented, the meaning of overexpression mutants becomes clearer.
Line 19: OsCSN2 must go in italics as you are referring to the gene here. I would recommend proofreading again all mutants, genes and proteins.
Section Headings 2.4 and 3.1: These headings could be improved for precision. My original suggestion was to incorporate more specific concepts into the section to better reflect the content and claims. I also recommend avoiding phrases like “Our study provides…” in headings. For example, the title for section 3.1 could be revised to: “OsCSN2 as a Potential Negative Regulator Under Blue Light” or something like that.
I would also recommend providing supplemental information and data, including the original datasets and raw images, and referencing this supplemental material appropriately within the manuscript.
